# Clinical Relevance of Vaginal and Endometrial Microbiome Investigation in Women with Repeated Implantation Failure and Recurrent Pregnancy Loss

**DOI:** 10.3390/ijms25010622

**Published:** 2024-01-03

**Authors:** Xushan Gao, Yvonne V. Louwers, Joop S. E. Laven, Sam Schoenmakers

**Affiliations:** 1Division of Reproductive Endocrinology and Infertility, Department of Obstetrics and Gynecology, Erasmus University Medical Center, 3015 CN Rotterdam, The Netherlands; 2Division of Obstetrics and Fetal Medicine, Department of Obstetrics and Gynecology, Erasmus University Medical Center, Wytemaweg 80, 3015 CN Rotterdam, The Netherlands

**Keywords:** vaginal microbiome, endometrial microbiome, repeated implantation failure, embryo implantation, recurrent miscarriage, recurrent pregnancy loss, pregnancy

## Abstract

Recent studies have investigated if and how the vaginal and endometrial microbiome might affect endometrial receptivity and reproductive health. Although there is no consensus on the existence of a core uterine microbiome yet, evidence shows that the dominance of *Lactobacillus* spp. in the female reproductive tract is generally associated with eubiosis and improved chances of successful implantation and an ongoing pregnancy. Conversely, vaginal and endometrial dysbiosis can cause local inflammation and an increase of pro-inflammatory cytokines, compromising the integrity and receptivity of the endometrial mucosa and potentially hampering successful embryonic implantation. This review provides a critical appraisal of the influence of the vaginal and endometrial microbiome as parts of the female reproductive tract on fertility outcomes, focusing on repeated implantation failure (RIF) and recurrent pregnancy loss (RPL). It seems that RIF as well as RPL are both associated with an increase in microbiome diversity and a loss of *Lactobacillus* dominance in the lower female reproductive system.

## 1. Introduction

Despite the fact that reproductive outcomes in subfertile couples undergoing assisted reproductive treatment (ART) have improved over the last decades [1], the underlying causes of adverse outcomes in early pregnancy are still mostly unknown. Current techniques cannot investigate or visualize essential developmental stages around and after human embryonic implantation. While the intra-uterine environment is safeguarding and allows early embryonic and placental development, it is inaccessible without jeopardizing the early pregnancy itself.

Chronologically, the essential steps to achieve a healthy pregnancy include ovulation, viable sperm cells, fertilization, blastocyst hatching, implantation, placental and embryonic development, and overall adaptation of the maternal immune system to support the early pregnancy. The emergence of ART provided insights into in-vitro fertilization, monitoring of the first embryonic developmental stages, and the possibility of embryo selection for embryo transfer. However, after embryo transfer, a successful pregnancy depends on the crucial embryo-maternal interaction during the period referred to as the black box of pregnancy [2].

The uterine microenvironment and its endometrial lining are crucial during the final pre-implantation stages for embryo apposition, followed by blastocyst hatching, embryo invasion, and endometrial receptivity, all necessary for the successful implantation of the embryo. Synchronization of the transformation into a receptive endometrial layer with the arrival and development of a competent blastocyst plays an important role in this early interaction and is essential for successful implantation. Importantly, the uterine microenvironment is thought to include microbiota and influence a local immune system, which in turn facilitates the embryonic-endometrium crosstalk and therefore acts as a co-determinant for implantation.

Over the last decades, the role of the microbiome in human health and disease, including pregnancy, has gained increased attention.

From the moment of birth, the developing neonatal immune system is initially exposed to microbes through vertical transmission, followed by continuous horizontal transmission from parents, peers, and the environment [3]. During vaginal delivery, the neonatal gut is colonized by the maternal vaginal and gut microbiomes. Offspring delivered via cesarean section are mainly exposed to the maternal skin microbiome, exhibiting a noteworthy reduction in microbiota diversity, structure, and composition compared to those born vaginally [4].

It is suggested that the developing neonatal and infant microbiome is needed for shaping both innate and adaptive immunity. In turn, the developing immune system also influences the shaping of the microbiome itself [5,6]. This interplay has extensively been studied in the context of gut microbiota [7].

The immune system consistently regulates microbes but reacts much stronger when they become invasive or exhibit other pathogen-related behaviors. The gut microbiota interacts with the host immune response and can induce an imbalance in cytokine levels, which could impact placentation and embryonic development, resulting in unexplained pregnancy loss [8,9]. Currently, research is also focusing on the possible role of a leaky gut in the pathophysiology of RPL [10]. Imbalances in the gut microbiome can lead to the production of endotoxins, for example, lipopolysaccharide (LPS). LPS is the main component of the outer membrane of gram-negative bacteria and serves as a marker of increased bacterial translocation from the gut lumen to the circulation via barrier dysfunction. However, in general, LPS is not regarded as inherently harmful. Instead, its endotoxic effects are mainly influenced by the timing of infection and the concentration of LPS, which are mediated through the activation of the innate immune system [11]. Tersigni et al. propose that, as a result of a leaky gut, LPS infiltrates the maternal circulation and activates the endometrial immune system, causing inflammation and leading to RPL [10]. We hypothesize a similar mechanism for the reproductive tract microbiota.

Although more research is needed to determine whether microbial dysbiosis is one of the explanatory variables of early pregnancy complications, current scientific evidence indicates that the association between a perturbed microbiome of the reproductive tract and implantation failure deserves more clinical consideration [12]. The vaginal microbiome is known to play a crucial role in vaginal health. Most research into reproductive outcome has shown that the dominant presence of *Lactobacillus* species within the female reproductive tract is associated with reproductive success, while adverse reproductive outcome seems to be associated with a non-dominant *Lactobacillus* environment and a generalized higher microbial diversity and richness [13,14,15,16]. Moreover, depletion of vaginal *Lactobacillus* spp. is associated with the presence of pro-inflammatory cytokines, which seem to be involved in reproductive failure [17,18]. The microbiota of the female reproductive tract is suggested to be a key player in the local uterine immune pathways operating during pregnancy [19]. This could imply that the consequences and effects of a dysbiotic microbiota on the reproductive outcome, resulting from bacterial invasion from the vagina into the uterine cavity, might be influenced by the local immune system’s response. If the immune system is excessively active during the bacterial invasion, this results in a pro-inflammatory state and potentially leads to either implantation failure or pregnancy loss.

Studies focusing on the influence of the microbiota of the female reproductive tract on endometrial receptivity and how this may impact pregnancy outcomes are increasing. However, many questions remain to be answered. In the current review, we aim to summarize current insights and research on the role of the vaginal and endometrial microbiome and the local immune responses in early pregnancy, focusing on RIF and RPL. We hypothesize an association between the success of early pregnancy, specifically implantation, and the composition of the female genital tract microbiome, immunological pathways, and local inflammation.

## 2. Role of Microbiota in Modulating Immune Tolerance during Early Pregnancy

### 2.1. Embryo Implantation

Embryo implantation is a highly organized process that involves a receptive endometrium and a competent blastocyst [20]. The endometrial lining needs to undergo functional remodeling to promote and allow invasion of the blastocyst. In the proliferative phase, the endometrial layer thickens due to the rising levels of estrogens, and after ovulation, the secretory phase starts when endometrial proliferation halts and more endometrial glands filled with glycogen develop under the influence of progesterone in preparation for implantation [21].

After blastocyst hatching, embryo implantation consists of three phases: (1) apposition, during which the blastocyst contacts the implantation site of the endometrium; (2) adhesion, in which trophectoderm cells adhere to the extruding pinopods of the epithelial layer of the endometrium; and (3) invasion, when extra-embryonic trophoblast cells penetrate the endometrial stroma by crossing the basement membrane and invading the endometrial epithelium [22].

To facilitate the endometrial-embryo cross talk, implantation is orchestrated by a synchronized molecular dialogue, directed from the embryo towards the maternal site and vice versa, mediated by cytokines, various growth factors, prostaglandins, matrix-degrading enzymes, and their inhibitors, and adhesion molecules [23]. Asynchrony in the cross talk between a functional blastocyst and a receptive endometrium, either by disruption of the dialogue or timing outside the window of implantation, can interfere with implantation. The pathophysiology of RIF can be attributed to either embryonic factors, such as genetic or chromosomal abnormalities, maternal factors, or a disturbed endometrial-embryo cross talk [24], whereas suboptimal implantation can result in an increased risk of miscarriage or RPL.

According to previous studies, endometrial impairment in selecting good-quality embryos and not rejecting incompetent embryos is suggested to play a role in women with RIF and RPL [25]. Current insights suggest that the endometrium generally accepts all embryos, followed by rejection of those of bad quality while embracing those of good quality [26,27]. According to a recent comprehensive analysis by Brosens et al., the endometrium includes specific checkpoints that must be cleared to reduce maternal investment in a failing pregnancy [26]. For instance, ‘implantation checkpoint’ refers to the implantation window where the human endometrium is prepared to activate both the mechanisms leading to (a) menstrual breakdown and (b) the formation of the decidua of pregnancy [26]. The endometrium of women with RPL is associated with an environment that lacks anti-inflammatory decidual cells and is highly conducive to embryo implantation but is prone to breakdown in pregnancy and is hostile to placental formation. This condition is also named ‘the implantation checkpoint failure’, wherein the endometrium fails to facilitate the development of a normal embryo and does not eliminate the abnormal embryos at implantation [26].

### 2.2. Immunology of Early Pregnancy

During pregnancy, the maternal immune system facilitates an immunosuppressive environment to prevent the rejection of the developing semi-allogenic fetus and placenta [28], which involves extensive modifications of both the adaptive and innate immune systems. These modifications provide a favorable microenvironment that promotes implantation and supports healthy placentation [23]. Uterine NK cells downregulate their cytotoxic activity to support this immunosuppressive environment for the fetus and placenta [29].

Around the time of blastocyst implantation, mild inflammation is therefore needed to accomplish endometrial receptivity, including the attraction of the developing blastocyst to the epithelium [23]. For this, the T-helper 1/T-helper 2 ratio changes, in which the pro-inflammatory Th1 cells become dominant, secreting cytokines like interleukin-2 (IL-2), IFN-γ, TNF-α, and TNF-β, which promote implantation of the developing embryo [30]. After implantation, the ratio of Th1/Th2 shifts back towards a Th2-dominated anti-inflammatory immune response, leading to the secretion of anti-inflammatory cytokines such as IL-4, IL-6, and IL-10, protecting the fetus and contributing to placental development [31].

### 2.3. The Role of the Reproductive Tract Microbiota in Early Pregnancy

The composition of the microbiota of the female reproductive tract serves as a proxy for implantation success, as studies show that the general presence of a *Lactobacillus* dominance microbiome is strongly associated with reproductive success. This might be explained by the fact that *Lactobacilli* thrive on a certain substrate in the vaginal and endometrial anaerobic environment found in epithelial secretions, which might be similarly crucial for the embryo’s early survival following transfer into the uterine cavity and during the pre-implantation period [32]. Both developing embryos as well as *Lactobacilli* need glycogen to survive, suggesting that a *Lactobacillus*-dominant micro-environment in the endometrium, if it exists at all, indicates an optimal environment for the embryo to implant [33].

*Lactobacilli* protect the local environment against pathogens by promoting tight junction protein expression [34] and the production of lactic acid, as well as some key bacterial metabolites like glycerophospholipids and benzopyran necessary for embryonic development and implantation [35]. Lactic acid lowers the local pH, which provides protection against pathogens and has stimulatory effects on the innate immune system when it is exposed to the gram-negative bacterial lipopolysaccharide [36]. The presence of lactic acid also allows for the breakdown of the endometrial layer, while glycerophospholipids are responsible for the synthesis of prostaglandins and play crucial roles in embryo implantation. Prostaglandins are involved in stimulating angiogenesis during early pregnancy and influence blastocyst transportation by regulating myometrium relaxation and contraction [37,38], and as such, a lack of prostaglandins is associated with lower pregnancy chances [35].

In summary, successful implantation and pregnancy are associated with a *Lactobacillus*-dominant (LD) microbiome in the female reproductive tract. In contrast, a non-*Lactobacillus* dominant (NLD) microbiome may contribute to an inflammatory response affecting embryo implantation [39]. Moreover, a lack of glycogen in both the endometrium and the vagina might reduce the number of *Lactobacillus* species and thereby prevent proper implantation.

## 3. Repeated Implantation Failure and Recurrent Pregnancy Loss

### 3.1. Definition, Prevalence, and Prognosis of Repeated Implantation Failure

Several definitions of RIF have been used over the years in the literature. RIF has been described as an iatrogenic condition resulting from three unsuccessful fresh in vitro fertilization (IVF) cycles [40], failing to achieve a pregnancy after three completed fresh IVF-cycles with their resulting embryo transfers [41], or failure after the transfer of at least four good-quality embryos within a minimum of three fresh or frozen cycles [42]. The prevalence varies depending on the definition of the RIF used. A recent evaluation showed that up to 15% of the 1221 women undergoing IVF treatment suffer from RIF, which is defined as the failure to obtain a clinical pregnancy after at least three embryo transfer attempts [43]. Based on the definition of three or more consecutive failed frozen cycles [42], another group of researchers found that the prevalence of RIF was 5% among 4.429 women (mean age 35.4 years) with up to three successive frozen euploid single blastocyst transfers [44].

The chance of an ongoing pregnancy resulting in a live birth after a diagnosis of RIF remains unclear. A recent study could not identify RIF as a clinical predictive parameter associated with a live birth [45]. However, the cumulative incidence of a subsequent live birth after RIF during a follow-up period of 5.5 years was 49% (95% CI 39–59%) among women under 39 years, with around 70% of these pregnancies resulting from fresh or frozen embryo transfers after IVF or IVF/ICSI [45]. The risk factors for RIF include maternal age, BMI, uterine abnormalities, chromosomal abnormalities, and lifestyle factors such as smoking and excessive alcohol consumption [46,47,48,49].

### 3.2. Definition, Prevalence, and Prognosis of Recurrent Pregnancy Loss

RPL includes primary and secondary RPL. Primary RPL characterizes women who never had an ongoing pregnancy resulting in a live birth. Secondary RPL refers to women who have had at least one live birth before the recurrent miscarriages. RPL is defined as two or more pregnancy losses before 24 weeks gestation, verified by ultrasound or the presence of chorionic villi on an endometrial biopsy. This includes pregnancies following spontaneous conception as well as after ART. The prevalence of RPL is 1–2% when defined as three consecutive pregnancy losses prior to 20 weeks of gestation [50]. The risk of subsequent pregnancy loss is associated with the number of pregnancy losses; the higher the number of pregnancy losses, the higher the risk of subsequent pregnancy loss. Egerup et al. showed that the number of consecutive early pregnancy losses after the last birth has a significant negative prognostic influence on women with secondary RPL [51]. The authors suggest that an ongoing pregnancy resulting in a live birth in women with secondary RPL ‘neutralizes’ the negative prognostic effect of previous pregnancy losses. In other words, only consecutive pregnancy losses should be counted in the definition of RPL [51]. Recently, Krog et al. analyzed 106 vaginal microbiota samples in women with primary or secondary RPL using the shot-gun sequencing technique [52]. They found a dominance of *Lactobacillus cripatus* in women with primary RPL and *Lactobacillus iners* with secondary RPL. Indeed, this implies that a previous successful pregnancy seems to influence the composition of the vaginal microbiota.

Risk factors for RPL include advancing maternal age, obesity, antiphospholipid syndrome, parental chromosomal abnormalities, uterine anomalies, smoking, diabetes mellitus, thyroid disease, and thrombophilia [44,53,54,55]. However, more than 50% of the RPL lacks a clear etiology.

### 3.3. The Role of Reproductive Tract Microbiota in RIF and RPL

Although the exact mechanism by which pathogenic bacteria contribute to RIF or RPL remains unclear, it is believed that pro-inflammatory immune responses of the host due to either a lack of *Lactobacilli*, the presence of pathogens in the uterine cavity, or a combination of both factors play a significant role. The interplay between the microbiota and the immune system in the reproductive tract is considered a key factor in understanding the underlying causes of RIF and RPL [56,57].

In addition, several mechanisms for the interaction between the microbiota and the endometrium have been suggested [56,57,58]. During the adhesion process of the embryo to the endometrial lining, inflammatory mediators are carefully regulated [57]. Benner et al. suggested that, during implantation, the immune system is activated via pattern recognition receptors present on the endometrial epithelial cells. Usually, the presence of commensal bacteria can induce similar interaction between the host and microbes, fostering the tolerance of commensals. On the other hand, the presence of pathogenic bacteria and the depletion of commensal bacteria, along with their beneficial molecules, could potentially compromise the integrity of the endometrial mucosal barrier. This interference may alter the mucosal T-cell balance, leading to dysregulation of cytokine levels in immune cell activation and negatively affecting the local immune environment [58]. As a consequence, this cascade of events might hinder the proper invasion of the embryo [56].

In summary, subsequent activation of systemic and local tissue immune responses during pregnancy can lead to inflammation and potential damage to placental tissue and compromise placental function. Vaginal microbiota in bacterial vaginosis are known to interact with and activate vaginal dendritic cells, potentially causing overactivation of the immune system, which has been associated with adverse reproductive outcomes [59].

### 3.4. Repeated Implantation Failure

#### 3.4.1. Vaginal Microbiome

Various factors can lead to the dysbiotic state of the vaginal microbiome, such as infections, usage of vaginal douches [60], certain diets [61], as well as prolonged use of antibiotics or other medications that eliminate bacteria [62]. The vaginal microbiome in women with RIF showed significantly higher microbial diversity compared to control groups [35]. Several independent studies show high abundances of genera associated with bacterial vaginosis, such as *Gardnerella*, *Prevotella*, *Atopobium*, *Megasphaera*, *Burkholderia*, and *Sneathia*, and low abundances of *Lactobacilli* in the RIF population compared to controls [13,35,63,64,65,66]. At the species level, women with RIF have a higher relative abundance of *L. helveticus*, while *L. iners*, *Lactobacillus jensenii*, *Lactobacillus gasseri*, and *Lactobacillus agalactiae* were more prevalent in women without RIF [63]. However, the presence of *L. iners* in women with RIF and unexplained infertility suggests an unfavorable role of this species in fertility [67]. Future studies should include a species-level analysis to identify the specific roles of the *Lactobacillus* species in reproductive outcomes.

#### 3.4.2. Uterine Microbiome

In contrast to the vaginal microenvironment, if any at all, the uterine cavity maintains an extremely low microbial population and harbors 10,000 times fewer bacteria [68,69]. Due to this low biomass, no consensus exists on the composition of healthy endometrial microbiota or even the existence of a core microbiome due to the inaccessibility of the uterine cavity without the risk of contamination [69,70]. A recent study showed that in 141 women with RIF, 121 endometrial samples were non-*Lactobacillus* dominant with an increased presence of *Streptococcus*, *Staphylococcus*, *Neisseria*, and *Klebsiella* [71]. These pathogens are known to compromise the integrity of the endometrial epithelial, which can lead to failed implantation [72]. Also, the prevalence of *Bacteroides* species in the endometrium of non-pregnant women has been linked with RIF [73] A retrospective study has shown a prevalence of up to 66% of chronic endometritis in women with RIF diagnosed by hysteroscopy and biopsy [72]. Another study reported that *Lactobacillus* was less abundant in chronic endometritis, whereas *Dialister*, *Bifidobacterium*, *Prevotella*, *Gardnerella*, and *Anaerococcus* were more abundant in the endometrial microbiota of women with chronic endometritis [74]. However, microbial imbalances in chronic endometritis might also induce an altered immune response, which can cause a decrease in the endometrium’s receptivity, leading to RIF [75]. A dysbiotic vaginal microbiome and its associated pro-inflammatory responses may damage the cervical epithelial barrier, allowing bacterial translocation to the endometrium, which can lead to a local infectious state interfering with early pregnancy [76]. See Table 1 for a summary of the vaginal and endometrial microbiome in women with RIF.

Although an overactive maternal immune system is deemed to be disruptive for implantation and derangements within the endometrial immune environment are suggested to be involved in women with RPL, a comprehensive assessment of the biological basis of the immunology of RIF and RPL is beyond the scope of this review. We aim to discuss the most investigated immunological factors associated with RIF and RPL.

#### 3.4.3. Molecular Immunological Characteristics of RIF

##### Natural Killer Cells

Natural killer (NK) cells, a subtype of cytotoxic lymphocytes, are key players in innate immunity and play an important role at the fetomaternal interface, making up around 70% of all immune cells [79]. Uterine NK (uNK) cells present in the uterus are involved in the process of successful placentation via angiogenesis and remodeling of the uteroplacental spiral arteries [80,81]. Additionally, NK cells contribute significantly to the establishment and maintenance of pregnancy through interactions with trophoblast cells [82], assisted by the production of cytokines, growth factors, and various chemokines. These factors all have an immunoinhibitory effect and are believed to promote a tolerogenic environment for the embryo [80,81]. Implantation failures have been associated with elevated cytotoxic NK cells in the decidua, a decrease in regulatory NK cells, and dysregulated cytokine production [83,84,85]. Dysregulation of NK cell cytokine production, particularly pro-inflammatory cytokines IFN-γ, and TNF-α, seems to be strongly associated with the immunopathology of RIF and RPL [84]. Activation of IFN-γ and TNF-α not only has inhibitory effects on blastocyst development but may also damage the tight junctions of the endocervical epithelial barrier, leading to a higher susceptibility to infection and inflammation [84].

Furthermore, with a dysregulated immune system, the interaction between uNK cells and the human leukocyte antigens (HLAs) expressed by trophoblasts, which is crucial in preventing trophoblast rejection by the maternal endometrium, becomes ineffective [86].

##### T Lymphocytes

Balances in diverse T-helper lymphocytes, i.e., Th1, Th2, Th17, and regulatory T cells (Treg), are important to mediate maternal tolerance to the semi-allogenic fetus [87]. The roles of Th1 and Th2 have been discussed before. After implantation, Treg cells are important in the maternal immune tolerance towards the fetus in the first and second trimesters through the production of the immunosuppressive cytokines IL-10 and TGF- β [88]. Th17 cells produce the regulatory cytokine IL-17, which is believed to be essential for the maintenance of a successful pregnancy [89]. An imbalance in the immunological Th17/Treg ratio is thought to be one of the contributing factors to the occurrence of RIF [90].

#### 3.4.4. Reproductive Microbiome and Immunology in RIF

A recent study focused on the immunological markers of endometrial samples in relation to embryo implantation [78]. Endometrial biopsy samples were obtained from 26 women with RIF undergoing IVF treatment and were divided into a non-*Lactobacillus* dominant (NLD) and *Lactobacillus*-dominated (LD) microbiota (<90% and ≥90% *Lactobacillus*, respectively) [78]. The authors observed a significantly positive correlation between the *Lactobacillus*-dominant group (eubiosis) and the increased levels of cytokines that promote anti-inflammatory characteristics in tissues such as IL-10 and IGF-1 [78]. Moreover, a significantly negative correlation was found between *Lactobacillus* abundance and the pro-inflammatory cytokines IL-1β and IL-6 [78]. The inflammatory state could be explained by the depletion of *Lactobacillus*, the presence of pathogens, or a combination of both. In the samples from the group where *Lactobacillus* was not dominant, a higher level of IL-6 was associated with the presence of anaerobe microbiota such as *Gardnerella* and *Streptococcus* [78]. This is in line with another study focusing on uterine immunological changes, which found higher expression levels of IL-6 in women with RIF [91]. In contrast, a study in women with RIF reported lower levels of IL-6 in the endometrial [92] stroma. Based on the discrepancies observed, no definite conclusion can be drawn about the role of the ILs and endometrial microbiota in RIF pathophysiology.

Recent research has shown that in women with RIF, a dysbiotic vaginal microbiome (<90% *Lactobacillus*) was associated with increased serum levels of pro-inflammatory cytokines, specifically TNF-α and IFN-γ [93]. The presence of *Burkholderia*, *Chryseobacterium*, *Enterococcus*, *Escherichia*, *Gemella*, *Herbaspirillum*, *Negativicoccus*, and *Staphylococcus* was linked to an elevated TNF-α/IL-6 ratio. While the abundance of *Aerococcus*, *Burkholderia*, *Escherichia*, *Herbaspirillum*, *Megasphera*, *Pseudomonas*, *Ralstonia*, and *Staphylococcus* was associated with an increased IFN-γ/IL-10 ratio [93]. These findings further suggest a correlation between dysbiotic microbiota and an immune system in favor of inflammation.

### 3.5. Recurrent Pregnancy Loss

#### 3.5.1. Vaginal Microbiome

A dominance of *Lactobacillus* spp. is found to play a protective role in the vaginal microbiota in early pregnancy, while vaginal dysbiosis is associated with RPL in ART patients [30,94,95,96]. The vaginal microbiota from women with depleted levels of *Lactobacillus* spp. displayed increased diversity of potential pathogens such as *Streptococcus*, *Prevotella*, *Ureaplasma*, *Peptoniphilus*, *Dialaster*, *Megasphaera*, *Sneathia sanguinegen*, *Gardnerella*, and *Atopobium* [17,30,75,94,95,96,97,98,99,100,101]. Several of these microbes are classified as part of the community state type (CST) IV, which is associated with a dysbiotic vaginal microbiota [30].

A recent study did not find a difference in vaginal microbiota, including *L. iners*, *G. vaginalis*, *Atopobium vaginae*, and *Bifidobacterium breve*, between women with RPL (*n* = 88) and healthy controls (*n* = 17) in a Japanese population [57], although this study is limited by its small sample size. Another recent study, comparing the vaginal microbiota of women with a normal pregnancy that ended through an induced abortion and women with unexplained RPL, showed that the latter group had an increased incidence of *Pseudomonas*, *Roseburia*, *Collinsella aerofaciens*, and *Arthrobacter* [94]. Therefore, analyses of the vaginal microbiome could act as a starting point for diagnostic investigations in RPL, especially in women undergoing assisted reproductive technology (ART) treatment.

Several studies have presented associations between the specific vaginal presence of *Atopobium* and RPL. One study found a high abundance of *Atopobium* in the vaginal microbiome samples of women with RPL as compared to controls [97], and another study reports that a relative abundance of >0.01% *Atopobium* could be used as a potential microbial biomarker to predict spontaneous miscarriages in the first trimester [102]. The potential immune-invasive role of *Atopobium* could be involved in disrupting the physicochemical barrier of the vaginal mucosa, thereby causing the release of pro-inflammatory cytokines and invasion of other anaerobes [102]. However, more research is needed on how the bacterial load of *Atopobium* affects pregnancy loss before it can be implemented as a diagnostic test.

#### 3.5.2. Uterine Microbiome

Analyses in the microbiota of endometrial tissue and uterine lavage fluid showed significantly different microbes as compared with the lower genital tract, with relatively higher abundances of *Acinetobacter*, *Anaerobacillus*, *Erysipelothrix*, *Bacillus*, and *Hydrogenophilus* spp. in the RPL group compared to controls [103]. Another study collected endometrial samples by endometrial curette, which showed the dominance of *L. iners* instead of *L. crispatus* in the RPL group compared with controls [99]. *L. iners* has been reported to be associated with endometrial dysbiosis and adverse reproductive outcomes, including subfertility [104] and a history of pregnancy loss [105]. Recently, it was shown that 22 different significant taxa exist in endometrial fluid versus endometrial tissue, indicating that uterine lavage may not entirely reflect the microbial composition within endometrial tissue [106]. Endometrial samples from women with RPL were found to have a higher bacterial diversity and richness with species such as *Aliihoeflea*, *Acinetobacter*, *Staphylococcus*, and *Serratia* [106]. Another prospective study showed a relative dominance of *Ureaplasma* species in the endometrial microbiome tissue as an independent risk factor for a subsequent miscarriage after an embryo transfer with an euploid karyotype [107]. Furthermore, the composition of the endometrial microbiota during a menstrual cycle appears significantly different between women with RPL and controls [77]. Women with recurrent pregnancy loss (RPL) have a distinct endometrial microbiome characterized by maintained diversity and richness throughout the menstrual cycle, while in the control group, the microbiota decreases in diversity near ovulation and remains stable during the luteal phase [77]. This observation serves as a distinguishing factor between women who experience RPL and those who do not miscarry.

To summarize, the decreased abundance of *Lactobacillus* spp. and increased richness and diversity of the endometrial microbiome in women with RPL appear to be associated with miscarriages.

See Table 2 for a summary of the vaginal and endometrial microbiome in women with RPL.

#### 3.5.3. Molecular Immunological Characteristics of RPL

##### NK Cells

Peripheral NK (pNK) cells before pregnancy seem significantly increased in women with RPL compared with controls, and this also differs between women with primary and secondary RPL [108,109,110]. Women experiencing primary RPL demonstrate significantly higher levels of pNK cells when compared to women who have had a previous live birth and nulliparous control women [111]. However, there is no consensus on whether the increased level of NK cells represents cytotoxicity in women with RPL before pregnancy [112,113]. Therefore, testing preconceptional pNK cell activity to predict subsequent miscarriage in women with RPL is not recommended [55].

##### Cytokines

In addition, increased levels of NK cells, increased plasma levels of tumor necrosis factor-α (TNF-α), interferon-gamma (IFN-γ), and IL-6 seem to be associated with RPL [109,114]. TNF-α promotes trophoblast invasion and induces vascular endothelial growth factor (VEGF) important for placental development [115]. TNF-α and IFN-γ are Th1 cytokines commonly present in acute and chronic inflammatory conditions. Their presence poses a potential risk, as they may cause direct damage to the placenta and fetus or indirectly activate cytotoxic cells, including NK or T cells [116]. However, one study did not find differences in the levels of IFN-γ and TNF-α between women with RPL and a successfully completed pregnancy compared to those who experienced another miscarriage [117].

##### T lymphocytes

An imbalance between Th17 cells and Treg cells also plays a role in RPL [87,118]. In the event of a bacterial invasion, Th17 is activated and contributes to inflammation by secreting cytokines such as IL-17A, IL-22, and IL-26 [118]. A higher Th17/Treg cell ratio was found in studies with unexplained RPL compared to fertile controls [119,120]. These findings suggest that an imbalance in Th1/Th2 and Th1/Treg cell ratios might have a negative impact on implantation and the maintenance of pregnancy.

#### 3.5.4. Reproductive Tract Microbiome and Immunology in RPL

Uterine and peripheral NK cells are considered members of the innate immune system and play an active role in pregnancy support. Chemokines, cytokines, growth factors, and angiogenic factors regulate trophoblast migration and promote placental growth. NK cells also play an important immunoregulatory role in the defense against infections [55]. In women with RPL, high levels of peripheral NK cells and an increased abundance of *G. vaginalis* and gram-negative anaerobes are observed in the vaginal microbiota, suggesting a potential link between the microbial composition, local inflammation, alterations in immune parameters, and the occurrence of miscarriage [95].

The pro-inflammatory cytokine IL-2, primarily produced by Th1 cells, and the anti-inflammatory IL-10, mainly secreted by Th2 cells, both play important roles in promoting successful implantation [87,121]. Studies focusing on the vaginal compartment found greater levels of IL-2 and lower levels of IL-10 in women with unexplained RPL compared to those with viable pregnancies [122]. Earlier research has discovered a connection between RPL and the down-regulation of IL-10 [123]. However, the mechanisms responsible for the reduced production of IL-10 at the maternal-fetal interface remain unclear. Pro-inflammatory cytokines such as IL-6, IL-1β, and TNF-α seem to be associated with a decrease in *Lactobacillus* and an increase in *Prevotella* and *Streptococcus* in women with euploid miscarriages [17].

Endometrial samples from women with RPL were found to have a higher bacterial diversity and richness and were associated with decreased IFN-γ and IL-6 pro-inflammatory cytokine concentrations [103]. Also, statistically significant correlations between *Aliihoeflea* and IL-17A, *Acinetobacter* and IFN-γ, *Serratia* and TNF-α, and *Staphylococcus* and *Serratia* and IL-6 were found [103].

## 4. Summary

This review provides an overview of the interaction of the reproductive tract microbiota, focusing on the vagina and uterine cavity, and the local immune responses in women experiencing RIF and RPL. Importantly, there seems to be a detectable difference in vaginal microbiome composition between healthy controls and women with RIF or RPL before conception. Women with RIF or RPL fail to attenuate their endometrial microbiome diversity and richness throughout the menstrual cycle [77].

A *Lactobacillus*-dominant vaginal microbiota seems to be beneficial for embryo implantation, whereas a non-*Lactobacillus*-dominant microbiota seems to impair implantation success. The pathogens identified in the dysbiotic vaginal microbiota of women with RIF were also found to be present in the dysbiotic vaginal microbiota of women with RPL. Among these pathogens, *Gardnerella*, *Prevotella*, *Atopobium*, *Megasphaera*, and *Sneathia* were observed to have high abundances. In contrast, the increased abundance of genera found in the endometrial microbiota is different among women with RIF and RPL.

Studies have discovered positive correlations between a higher abundance of *Lactobacillus* and anti-inflammatory cytokines, while a negative association has been observed between *Lactobacillus* and pro-inflammatory cytokines in the endometrial microbiota of women with RIF [78,91]. Additionally, pro-inflammatory cytokines have been positively linked to *Gardnerella* and *Streptococcus* [78].

Moreover, the innate and adaptive immune systems seem to be dysregulated in women with RPL. Predominant Th1 and Th17 immunity and decreased Th2 and Treg cells are associated with the RPL of fetuses with a normal karyotype [124]. Failures to maintain adequate Th1/Th2 and Th17/Treg cell balance have been associated with RIF and RPL [119,120].

It seems that, compared to eubiosis (Figure 1), vaginal and/or endometrial dysbiosis is associated with RIF (Figure 2) and RPL (Figure 3), and that in both, similar mechanisms are involved. Moreover, inflammation and dysregulated activation of the immune system seem to affect the integrity of the endometrial mucosa, leading to RIF, and subsequently interfere with the receptivity of the endometrium and the processes of implantation and placentation, leading to RPL.

The reproductive microbiome does not necessarily have to be limited to the reproductive tract, since recent evidence suggests that an imbalance of gut microbes, or gut microbial dysbiosis, can also lead to adverse pregnancy outcomes [8,125]. A study found a correlation between decreased gut microbiome diversity and an increased ratio of *Firmicutes* to *Bacteriodetes* in women with first-trimester miscarriages, indicating that the pro-inflammatory effect of the gut microbiome is probably caused by holistic dysbiosis rather than the dominance of a particular pathogen [125].

The gut microbiome is a crucial factor for regulating and shaping the immune response, and gut microbial dysbiosis can trigger inflammation [9]. Also, the Th1/Th17-mediated pro-inflammatory state in unexplained RPL is associated with gut dysbiosis [125]. The extent of the impact of the gut microbiota on reproductive organs is being increasingly recognized, revealing a greater influence than was previously understood. A pathological state of the intestinal system, the so-called leaky gut, is one in which imbalances in the gut microbiota can lead to the production of toxins that infiltrate the maternal circulation and activate the endometrial immune system, leading to inflammation and RPL [10]. The gut microbiome also affects hormonal homeostasis and the coagulation system, which are involved in implantation [67,126]. In the presence of increased abundances of Gram-negative bacteria in the gut microbiota, causing inflammation and the systematic release of pro-inflammatory cytokines, the secretion of luteinizing hormone (LH) and follicle-stimulating hormone (FSH) could be impaired, resulting in subfertility [127]. Future research will help to clarify the relationship between gut microbiota and reproductive outcomes.

## 5. Challenges and Future Perspectives

Microbiome studies are complicated by the diversity in the execution of analytical steps, which can cause heterogeneity due to differences in local protocols, sampling methods, primers used, sequencing techniques, and bioinformatics pipelines, making comparisons between studies difficult, especially across different ethnicities. To obtain comprehensive insights into the role of *Lactobacillus* spp. in reproductive health, studies should incorporate a species-level analysis of the vaginal and uterine microbiota, as microbiota dominated by *Lactobacillus* may not always indicate the presence of the beneficial *L. crispatus*, but rather an intermediate state towards a dysbiotic condition, often associated with the presence of *L. iners* or *L. gasseri* [128].

Furthermore, conducting species-level analysis allows for better comparisons between the studies conducted in different ethnicities. *Lactobacilli* species and their proportions are different among different ethnicities. Black African and African-American women carry more often *L. iners* compared to Caucasian women, who carry more likely *L. crispatus* [129].

Also, the obstetric history is of importance since the dominance of *L. crispatus* in the vaginal microbiota could change to *L. iners* in Caucasian women who gave birth [52]. Indeed, one study found that the abundance of *L. crispatus* is negatively correlated with the number of previous deliveries. In other words, a nullipara has a two-times higher mean relative abundance of *L. crispatus* than women with one prior delivery (58% vs. 26%) [130]. Future studies should take into account the obstetric history as well as the number of previous pregnancies at the time of sample collection.

Finally, during natural conception, the microbial content present in sperm may enter the uterine cavity and potentially interact with the local microbiota of the endometrial environment [19]. The role of the seminal microbiome in sperm health lacks consensus. Some studies indicate that *Lactobacillus*, commonly abundant in semen, is often found to be less abundant in men with lower seminal quality [131,132]. Others found conflicting results and showed an increase in *Lactobacillus* in patients with azoospermia [133]. *Prevotella*, on the other hand, consistently correlates with reduced sperm quality, primarily affecting sperm motility [131,132]. Although more research is needed to investigate whether the dysbiotic vaginal microbiota in women, which is linked to unfavorable fertility outcomes, could be reflected in the seminal microbiome of their partners,

Last but not least, to understand the molecular mechanisms simultaneously interacting around the timing of implantation, the use of different ex vivo-omics could offer more insights. Combining analyses and integrating techniques including metabolomics, transcriptomics, proteomics, and metagenomics for a single sample or condition provides a more comprehensive understanding of the taxonomic composition, local gene expression, and local molecular mechanisms involved in RIF or RPL.

A metatranscriptomics approach in microbiome research could be used to uncover specific information regarding transcriptionally active bacteria. Metatranscriptomics of chorionic villi tissue and decidua of women with RPL have shown upregulation of pro-inflammatory genes and dysregulation of genes involved in angiogenesis. These results indicate a link between inflammation and abnormal placental development in women with RPL [134,135]. Another study reported dysregulation of genes involved in the extracellular matrix and its remodeling, crucial for proper implantation and placentation during trophoblast invasion [136]. Combining these omics data can lead to a better understanding of the relationship between taxonomic composition and local gene expression patterns.

In the decidual tissue of women with RPL, proteomics showed overexpression of proteins related to the inhibition of decidual cell growth and oxidative stress pathways. Oxidative stress, caused by chronic inflammation, can interfere with mitochondrial function and cause DNA damage, thereby affecting embryonic development [137].

Future research should also focus on the use of blastoids, endometrial, and placental organoid models to mimic the physiological microbial-immunological and endometrial-embryo cross talk which provides the opportunity to study the local interaction with bacterial products, gene expression, and immunological pathways and non-invasively offer new insights into the pathophysiology of RIF and RPL.

## 6. Conclusions

In conclusion, the current review provides an updated overview of the vaginal and endometrial bacterial communities and their interaction with the local immune system in RIF and RPL. A state of microbiota dysbiosis with an overabundance of pathogenic species or the absence of *Lactobacillus*-dominated vaginal and endometrial microbiomes might trigger inflammation, hinder the process of embryonic implantation, and interfere with early pregnancy. To attain a better understanding of the role of *Lactobacillus* spp. and other microbes within the reproductive tract microbiota in reproductive health and disease, it is crucial to include species-level analyses of the vaginal and uterine microbiota in studies. This approach not only provides a more detailed examination but also facilitates more accurate comparisons among different studies conducted in diverse ethnic and obstetric populations. Future research should also be conducted in larger sample sizes and incorporate omics technologies and organoids to further enhance our understanding of reproductive health and disease.

## Figures and Tables

**Figure 1 ijms-25-00622-f001:**
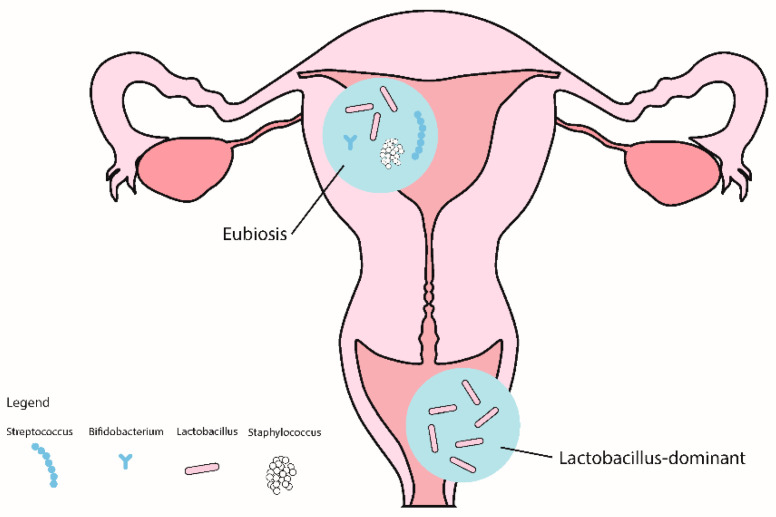
Eubiotic vaginal and endrometrial microbiota.

**Figure 2 ijms-25-00622-f002:**
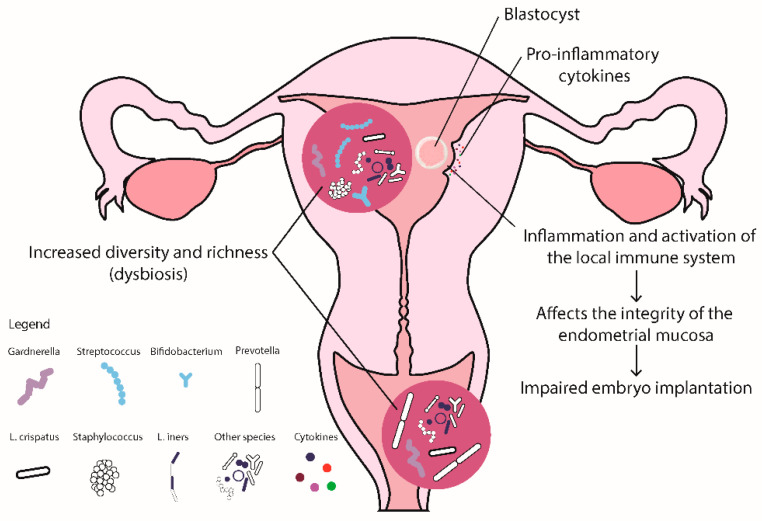
Alterations of the vaginal and endometrial microbiota in women with RIF.

**Figure 3 ijms-25-00622-f003:**
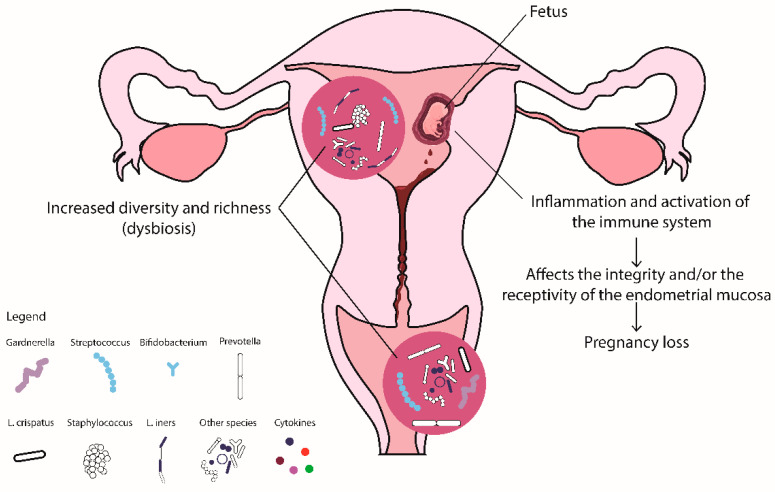
Alterations of the vaginal and endometrial microbiota in women with RPL.

**Table 1 ijms-25-00622-t001:** Summary of the vaginal and endometrial microbiome in women with RIF.

First Author (Year)	Study Group	Country	Specimen Type	Composition: Phylum/Genus/Species	Conclusion
**VAGINA**					
Kitaya et al. (2019)	RIF patients (*n* = 28)	Japan	Vaginal swab	*Lactobacillus* (>90%) dominance in the RIF group.	There were no significant differences in the community composition of the vaginal microbiome between the RIF group and the control group.
	Control *n* = 18 (undergoing first IVF attempt)				
Ichiyama et al. (2021) [65]	Women with RIF (*n* = 145)	Japan	Vaginal swabs	*Atopobium*, *Megasphaera*, *Gardneralla*, *Prevotella* ↑*	The vaginal microbiota of the RIF group had lower levels of *Lactobacillus* than in the control group.
	Healthy controls (*n* = 21)			*Lactobacillus* ↓*	The dysbiotic microbiota from the RIF group consisted of significantly higher levels of 4 other genera.
Diaz-Martinez et al. (2021) [63]	IVF patients with a RIF history (*n* = 23)	Spain	Vaginal swabs	*Genus*	There was no significant difference in *Lactobacillus* between the groups with RIF in history and without.
				*Streptococcus* ↓	
	IVF patients without RIF in history (*n* = 25)			*Preovotella* spp., *Ureaplasma* spp. and *Dialister* spp. ↑	
				*Species*	
				*L. helveticus* ↑	
Patel et al. (2022) [67]	Women with RIF (*n* = 10), unexplained infertility (*n* = 10) and healthy controls (*n* = 10)	India	Vaginal swabs	The dominance of *Lactobacillaceae* in all the groups *Lactobacillus iners* AB-1 dominance in the unexplained infertility group.	There was no difference in lactobacillus between the group with RIF, unexplained infertility, and healthy controls.
				*Leptotrichia* and *Sneathia* ↑ in control group *	
**ENDOMETRIUM**					
Kitaya et al. (2019)	RIF patients (*n* = 28)	Japan	Endometrial fluid	*Genus*	The endometrial fluid microbiota showed a significant difference in community composition between the RIF group and the control group. *Burkholderia* was only detected in the RIF group.
				*Lactobacillus* (>90%) dominance *Gardnerella* ↑	
	Control (infertile patients undergoing first IVF attempt *n* = 18)			*Burkholderia* ↑*	
Ichiyama et al. (2021) [65]	Women with RIF (*n* = 145)	Japan	Endometrial tissue with Pipette	*Atopobium*, *Megasphaera*, *Gardnerella*, *Prevotella*, *Schlegelella*, *Delftia*, *Burkholderia*, *Sphingobacterium*, *Dietzia, Enterococcus*, *Micrococcus*, *Ralstonia*, *Leucobacter*, and *Hydrogenophaga* ↑*	The dysbiotic microbiota from the RIF group consisted of significantly higher levels of 14 other genera.
	Healthy controls (*n* = 21)				
Diaz-Martinez et al. (2021) [63]	Patients with a RIF history (*n* = 23)	Spain	Transcervical endometrial tissue (Tao Brush IMC Endometrial Sampler)	*Genus*	The RIF group had significantly different compositions of the endometrial fluid microbiota. A higher abundance of *Prevotella* was detected in RIF patients.
				*Prevotella* and *Sneathia amnii* ↑*	
	and without RIF in history (*n* = 25)			*Species*	
				*L. helveticus* ↑*	
Keburiya et al. (2022)	Women with RIF (*n* = 91)	Russia	Endometrial tissue (Embryo transfer catheter tips)	Obligate anaerobes (*streptococci*, *enterobacteria*) were found in the RIF group but with low concentrations and were insignificant.	*G. Vaginalis* was significantly higher in the naïve IVF patients but did not have a significant impact on embryo implantation. There was no significant difference in *Lactobacillus* between the RIF group and naïve IVF patients.
	Women with first IVF attempt (*n* = 39)				
Chen et al. (2022) [75]	Women with RIF (*n* = 75)	China	Transcervical endometrial tissue	*Phylum*	There was a significant difference in community composition at the genus level of the RIF group compared to the control group. The dominance of *Sphingomonas* was found in the endometrial microbiota of the RIF group
				Dominance of *Firmicutes*, Proteobacteria, Bacteroidetes, Cyanobacteria, and Actinobacteria	
	Healthy controls (*n* = 36)			*Genus*	
				*Sphingomonas*, *Brevundimonas*, *DMER64*,	
				*Methylobacterium*, *Rhodoferax*, *Caulobacter* ↑*	
				*Lactobacillus*, *Pseudomonas*, *Delftia* ↓*	
Vomstein et al. (2022) [77]	Women with RIF (*n* = 20)	Finland	Transcervical endometrial tissue	*Phylum*	Women with RIF fail to attenuate their endometrial microbiome diversity and richness throughout the menstrual cycle
				*Firmicutes* further decrease after ovulation.	
	Healthy controls (*n* = 10)			Proteobacteria increase after ovulation.	
				Bacteroidetes peak in the follicular phase.	
				*Gardnerella* and *Dialister* were increased around ovulation	
Cela et al. (2022) [78]	Women with RIF (*n* = 26)	Italy	Transcervical endometrial tissue	*Genus*	*Lactobacillus* abundance is found to be negatively correlated with anti-inflammatory cytokines and positively correlated with pro-inflammatory cytokines.
	With LD (*n* = 13)			*Lactobacillus* ↓*	
	And NLDM (*n* = 13)			*Gardnerella*, *Streptococcus* and *Bifidobacterium* ↑	
Zou et al. (2023) [71]	Women with (*n* = 141)	China	Transcervical endometrial tissue	*Genus*	Diverse pathogenic bacteria were found in high abundance in the endometrial microbiota of women with RIF.
				*Streptococcus*, *Staphylococcus*, *Neisseria*, and *Klebsiella* ↑	
Iwami et al. (2023)	RIF patients with endometrial microbiota data (*n* = 131): *n* = 30 (23%) with abnormal microbiota.	Japan	Endometrial tissue with pipelle	*Genus*	Treatment with probiotics in women with RIF and abnormal endometrial microbiota might improve IVF outcomes.
				*Streptococcus, Gardnerella, Atopobium* and *Bifidobacterium* ↑	
Lozano et al. (2023)	Women with RIF (*n* = 27)	Spain	Transcervical endometrial tissue (Tao Brush IUMC Endometrial sampler)	*Genus*	*Lactobacillus* is adversely associated with pathogenic bacteria (*Prevotella, Dialister*, and *Streptococcus),* and this dysbiosis might be the cause of an increased risk of implantation failure.
				*Lactobacillus* ↑* (control 97.96% vs. RIF 92.27%)	
	Women without RIF (*n* = 18)			*Prevotella* ↑* (control 0.00% vs. RIF 2.19%)	
				*Dialister* ↑* (control 0.06% vs. RIF 0.15%)	
				*Streptococcus* ↑* (control 0.05% vs. RIF 0.18%)	

↑ = increased abundance of the bacteria, ↓ = decreased abundance of the bacteria * = significant difference between the study group and the control group. Abbreviations: LD: *Lactobacillus*-dominant; NLDM: Non- *Lactobacillus*-dominant microbiome.

**Table 2 ijms-25-00622-t002:** Summary of the vaginal and endometrial microbiome in women with RPL.

VAGINA					
First Author (Year)	Study Group	Country	Specimen Type	Composition: Phylum/Genus/Species	Conclusion
Kuon et al. (2017) [95]	Women with RPL (*n* = 243)	Germany	Vaginal swabs	*Genus*	RPL patients with elevated peripheral natural killer cells suffer more often from colonization by *G. vaginalis* and gram-negative anaerobes.
				*Group B Streptococcus* ↑	
				*Enterobacteriaceae* ↑	
				*Species*	
				*Gardnerella vaginalis* ↑	
Zhang et al. (2019) [96]	History of recurrent miscarriages (n = 10)	China	Vaginal swabs	*Phylum*	The dysbiotic microbiota from the RPL group consisted of an increased abundance of *Atopobium, Prevotella*, and *Streptococcus* and a decreased abundance of *Lactobacillus.*
				*Firmicutes* ↑*	
				*Actinobacteria* ↓*	
				*Bacteroidetes* ↓*	
	Healthy controls (n = 10)			*Genus*	
				*Atopobium, Prevotella* and *Streptococcus* ↑*	
				*Lactobacillus* ↓	
Al-Memar et al. (2020) [30]	Women with first term miscarriages (n = 64)	UK	Vaginal swabs	*Genus*	A reduced abundance of *Lactobacillus* was found in first-trimester miscarriages compared to women with viable pregnancies. Community State Type IV was more often found in miscarriages.
				*Streptococcus, Prevotella, Ureaplasma, Peptoniphilus,* and *Dialaster* ↑	
	Women with full term pregnancies (n = 83)			*Lactobacillus* spp. ↓	
Fan et al. (2020) [94]	Women with unexplained recurrent spontaneous abortion (n = 31)	China	Vaginal swabs	*Phylum*	The vaginal microbiota of women with unexplained RPL has much higher alpha diversity and a higher abundance of *Proteobacteria.*
				Proteobacteria ↑*	
	Normal pregnancy induced abortion (n = 27)			*Genus*	
				↑ *Gammaproteobacteria, Proteobacteria, Pseudomonas, Moraxella, Ruminococcus, Collinsella aerofaciens, Alteromonadaceae, Cellvibrio, Arthrobacter, Roseburia,* and *Micrococcaceae*	
Peuranpaa et al. (2022) [99]	Women with RPL (n = 47)	Finland	Vaginal swabs	*Species*	The vaginal microbiota of women with RPL has a higher abundance of *G. vaginalis* compared with healthy controls.
	Healthy controls (n = 39)			*Gardnerella vaginalis* ↑	
Caliskan et al. (2022)	Women with recurrent miscarriages (n = 25)	Turkey	Vaginal swabs	*Genus*	A decrease in *Lactobacillus* spp. And an increase in anaerobic microorganisms were found in the vaginal microbiota of women with RPL.
				*Lactobacillus* ↓	
	Healthy controls (n = 25)			*Enterobacterium* spp., *Eubacterium* spp.	
				*Megasphaera* spp. and *Sneathia* spp. ↑	
Jiao et al. (2022) [97]	History of recurrent miscarriages (n = 16)	China	Vaginal swabs	*Genus*	A decrease in *Lactobacillus* and an increase in the abundance of *Atopobium* and *Prevotella* were found in the vaginal microbiota of women with RPL.
				*Lactobacillus* ↓*	
	Healthy controls (n = 20)			*Gardnerella* ↓*	
				*Atopobium* ↑*	
Liu et al. (2022) [103]	Women with RPL (n = 25, with 50% chronic endometritis)	China	Vaginal swabs	*Genus*	*Lactobacillus* was the dominant genus in the RPL and control groups. No significant difference was found between the vaginal microbiota of the two groups.
				*Lactobacillus* (>90%)	
				*Bidifidobacterium* ↓	
	Healthy controls (n = 25, with 24% chronic endometritis)			*Gardnerella* spp. ↓	
				*Atopobium* ↑	
Ncib et al. (2022)	Women with unexplained RPL (*n* = 65)	Italia	Vaginal swabs	*Genus*	There is a high prevalence of aerobic vaginitis-causing bacteria in women with RPL (65%).
				*Enterococcus*,	
	Healthy controls with at least two live births and no miscarriages (*n* = 50)			*Staphylococcus*,	
				*Streptococcus* ↑	
Mori et al. (2023) [101]	Patients with unexplained RPL (*n* = 88)	Japan	Vaginal swabs	*Genus*	There is no difference in *Lactobacillus* and bacterial diversity between patients with RPL and live birth. *L. iners* dominance of the vaginal microbiota in the Japanese study population.
	Healthy controls with no history of miscarriage (*n* = 17)			Dominance of *Lactobacillus iners*, *Gardnerella vaginalis*, *Atopobium vaginae* and *Bifidobacterium breve.*	
**ENDOMETRIUM**					
**First Author (Year)**	**Study group**	**Country**	**Sampling**	**Composition: Phylum/** **Genus/Species**	**Conclusion**
Verstraelen et al. (2016)	19 women with RIF (*n* = 11), RPL (*n* = 7) or both (*n* = 1)	Belgium	Transcervical endometrial tissue (Tao Brush IUMC Endometrial sampler)	*Phylum*	90% of the women with either RPL or RIF had a uterine microbiota with a predominance of the phylum Bacteroidetes.
				Bacteroidetes ↑	
				Proteobacteria ↑	
				*Species*	
				*Bacteroides xylanisolvens, Bacteroides thetaiotaomicron, Bacteroides fragilis* ↑	
Peuranpaa et al. (2022) [99]	Women with RPL (*n* = 47)	Finland	Transcervical endometrial tissue	*Species*	*L. crispatus* is less abundant in women with RPL compared to controls (17.2% versus 45.6%). *G. vaginalis* is more abundant in the RPL group.
				*L. crispatus* ↓	
	Healthy controls (*n* = 39)			*L. iners dominance*	
				*Gardnerella vaginalis* ↑	
Liu et al. (2022) [103]	Women with RPL (*n* = 25, with 50% chronic endometritis)	China	Transcervical endometrial tissue	*Phylum*	A lower abundance *of Lactobacillus* spp. was found, and *Acinetobacter* spp. were predominant.
	Healthy controls (*n* = 25, with 24% chronic endometritis)			Proteobacteria ↑	
				*Genus*	
				*Lactobacillus* (<10%)	
				*Anaerobacillus* ↑*	
				*Erysipelothrix* ↑*	
				*Hydrogenophilus*↑*	
Shi et al. (2022) [107]	Patients with RPL (*n* = 63)	Japan	Transcervical endometrial tissue	*Genus*	A higher abundance of *Ureaplasma* species in the endometrium was associated with an increased risk of miscarriage with an euploid karyotype.
				*Ureaplasma* spp. ↑	

↑ = increased abundance of the bacteria, ↓ = decreased abundance of the bacteria, * = significant difference between the study group and the control group. Abbreviations: LD: *Lactobacillus*-dominant; NLDM: Non- *Lactobacillus*-dominant microbiome.

## Data Availability

Not applicable.

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
