# Peer review of "Clinical Relevance of Vaginal and Endometrial Microbiome Investigation in Women with Repeated Implantation Failure and Recurrent Pregnancy Loss"

_ijms, 2024, doi:10.3390/ijms25010622_

Round 1

Reviewer 1 Report (Previous Reviewer 1)

Comments and Suggestions for Authors

The revision improved the qualitiy of the manuscript. However, there are still some misleading statements.

It is misleading to state that the immune system distinguishes between healthy microbes and pathogens. Firstly, there is opportunistic pathogens. Secondly, rather the immune system holds microbes under control constantly, but react much stronger, when thy become invasive (and other pathogen-related behaviors). Similarly, LPS is not only carried by pathogens. It is a question of concentration and localization that makes LPS harmful.

Comments on the Quality of English Language

Please revise especially the newly added text. There are some spelling mistakes.

Author Response

Reviewer 2 Report (Previous Reviewer 2)

Comments and Suggestions for Authors

Author's justified the comments.

Author Response

Reviewer 3 Report (New Reviewer)

Comments and Suggestions for Authors

The authors presented an interesting study of the topic. Despite its extensiveness, the literature review remains incomplete. There are also minor stylistic and graphical flaws. I also have comments on the organization of the text.

Below are my comments:

1.       line 252 - I would not define the age of women >35 as advanced, even in the context of pregnancy - please correct it

2.       The figures are of very poor quality and their source is not indicated - this should be corrected

3.       The location of the figures in the text is very controversial. Discussion is the wrong place to put figures. I suggest that the authors separate a separate section in the text in which they present their concept, which is illustrated in the figures. Generally, I believe that in a review article, the discussion section is not necessary and the content of this part of the study can be described as a summary. All in all, a review of the available literature on a specific topic should lead to some conclusions. . I simply suggest titling the 5 sections of the text differently.

4.       The cited literature, although very extensive, is still incomplete - please supplement it with additional, interesting references from the PubMed database; e.g.

a.       Toson, B.; Simon, C.; Moreno, I. The Endometrial Microbiome and Its Impact on Human Conception. Int. J. Mol. Sci. 202223, 485. https://doi.org/10.3390/ijms23010485

b.       Lebedeva OP, Popov VN, Syromyatnikov MY, et al. Female reproductive tract microbiome and early miscarriages. APMIS. 2023;131(2):61-76. doi:10.1111/apm.13288

c.       Vitale SG, Ferrari F, Ciebiera M, et al. The Role of Genital Tract Microbiome in Fertility: A Systematic Review. Int J Mol Sci. 2021;23(1):180. Published 2021 Dec 24. doi:10.3390/ijms23010180

d.       Beckers KF, Sones JL. Maternal microbiome and the hypertensive disorder of pregnancy, preeclampsia. Am J Physiol Heart Circ Physiol. 2020;318(1):H1-H10. doi:10.1152/ajpheart.00469.2019

e.       …  These are just a few valuable studies that were missing from the presented review. There are many more in PubMed - please expand the review.

5.       The list of references is not formatted according to the journal's requirements - please correct it.

Comments on the Quality of English Language

The English language of this manuscript is acceptable.

Round 2

Reviewer 3 Report (New Reviewer)

Comments and Suggestions for Authors

 I stand by my previous comments.

Comments on the Quality of English Language

The English language of the manuscript is acceptable.

Author Response

This manuscript is a resubmission of an earlier submission. The following is a list of the peer review reports and author responses from that submission.

Round 1

Reviewer 1 Report

Comments and Suggestions for Authors

In this reviewing article submitted in the “International Journal of Molecular Sciences”, the authors describe the current state of the vaginal/endometrial microbiome with RIF and RPL. The topic is very interesting.

In my opinion, this article treats the subject superficially. It summarizes current data, but lack in giving a new idea or study questions (suggesting omics and organoids without context is not enough). As it was submitted to a “molecular” journal, it is barely discussed on a molecular level. Many reviews concerning the microbiome of the female reproductive tract are already existent and describe current studies in a similar way.

I appreciate the focus on RIF and RPL and the attempt to describe it separately. I think a deeper insight in molecular, immunological and microbial aspects would enrich the manuscript and strengthen its scientific impact.

·         Line 77; please avoid citing review articles in a review article. Most importantly, the reference does not contain the information stated in the sentence. Please use original literature

·         Paragraph 2.3; the authors mention the possible need of “certain substrates” for implantation, but also by lactobacilli. How can it be excluded that lactobacilli cause a shortage in glycogen for example.

·         I think it would be interesting for the topic that the authors include that the abundance of microbes is app. 10^4 less than in the vaginal microenvironment.

·         More is known about immunological changes in RIF and RPL. It is barely mentioned in the manuscript. Similarly, much more is known about immune implications in early pregnancy. The division of the immune components into pro- and anti-inflammatory does not describe it properly. There is at least type 1 (comparable to Th1), type 2 (comparable to Th2) and tolerogenic (comparable to Treg) immune reactions, but also local and systemical immune adaptations to early pregnancy. The authors already indicate the interactions of immune components and microbes, but hardly name it directly. Microbes shape immune responses and immune responses shape the microbial composition.

·         Why are the two tables in the supplementary data? – I would prefer showing them in the manuscript.

·         The quality of the figure is bad. I can barely read/evaluate it.

·         It would be interesting to not only describe the differences of RIF and RPL clinically, but also molecularly.

·         The used terms are misleading. Title: Vaginal and endometrial microbiome, 2.3 urogenital microbiome (although it is right, that lactobacilli also dominate the healthy urinary microbiome, it is not mentioned in this paragraph).

·         Might a healthy microbiome support implantation processes?

·         Some references seem to be lacking. Eg in line 233-234

·         In paragraph 3.3 “dysbiotic” microbes are described. What does that mean? The opposite to a healthy colonizers? They would also interact with PRR (which makes sense in terms of immune defense – thus the term “unfortunately” seems unprofessional in an immunological context). What is the difference between dysbiosis and infection?

Comments on the Quality of English Language

The language is appropriate. In some cases, shortening sentences might increase comprehension.

Reviewer 2 Report

Comments and Suggestions for Authors

The current manuscript "ijms-2607344" is focusing on how microbiota dysbiosis affect reproductive health. The article is very interesting and citied recent articles. Few minor changes are required to accept the manuscript.

1. Figures are not clearly visible and please extend the legend also.

2. It is nice to include healthy endometrial/vaginal microbiota levels in figure. 

3. Use same terminology for RPL 303 " Repeated pregnancy loss"